# BurstDeflicker: A Benchmark Dataset for Flicker Removal in Dynamic Scenes

**Lishen Qu**[1,3,5], **Zhihao Liu**[3], **Shihao Zhou**[1,3], **Yaqi Luo**[3]
**Jie Liang**[5], **Hui Zeng**[5], **Lei Zhang**[4,5], **Jufeng Yang**[1,2,3,*]

[1]Nankai International Advanced Research Institute (SHENZHEN·FUTIAN)
[2]Peng Cheng Laboratory     [3]College of Computer Science, Nankai University
[4]The Hong Kong Polytechnic University     [5]OPPO Research Institute
{qulishen, 2212602, zhoushihao96, 2323483}@mail.nankai.edu.cn
liang27jie@163.com,   cshzeng@gmail.com, cslzhang@comp.polyu.edu.hk
yangjufeng@nankai.edu.cn
https://github.com/qulishen/BurstDeflicker

## Abstract

Flicker artifacts in short-exposure images are caused by the interplay between the row-wise exposure mechanism of rolling shutter cameras and the temporal intensity variations of alternating current (AC)-powered lighting. These artifacts typically appear as non-uniform brightness distribution across the image, forming noticeable dark bands. Beyond compromising image quality, this structured noise also impacts high-level tasks, such as object detection and tracking, where reliable lighting is crucial. Despite the prevalence of flicker, the lack of a large-scale, realistic dataset has been a significant barrier to advancing research in flicker removal. To address this issue, we present *BurstDeflicker*, a scalable benchmark constructed using three complementary data acquisition strategies. First, we develop a Retinex-based synthesis pipeline that redefines the goal of flicker removal and enables controllable manipulation of key flicker-related attributes (e.g., intensity, area, and frequency), thereby facilitating the generation of diverse flicker patterns. Second, we capture 4,000 real-world flickering images from different scenes, which help the model better understand the spatial and temporal characteristics of real flicker artifacts and generalize more effectively to wild scenarios. Finally, due to the non-repeatable nature of dynamic scenes, we propose a green-screen method to incorporate motion into image pairs while preserving real flicker degradation. Comprehensive experiments demonstrate the effectiveness of our dataset and its potential to advance research in flicker removal.

## 1 Introduction

Flicker artifacts commonly arise in images captured under alternating current (AC)-powered light sources [1, 2, 3]. This phenomenon is especially prevalent in high-speed photography [4, 5, 6], high dynamic range (HDR) imaging [7, 8, 9], and slow-motion video recording [10, 11, 12], where short exposures are either required or frequently used. We provide a schematic overview of flicker formation as shown in Figure 1(a). Flicker artifacts arise primarily from two reasons. First, since the intensity of AC varies periodically, AC-powered light sources inherently exhibit periodic fluctuations in brightness [1, 3, 13]. Although these fluctuations are generally imperceptible to the human eye due to the persistence of vision [14], they become visible in images captured with short exposure times, which may sample only a narrow temporal slice of the flicker cycle. Second, most consumer-grade

---

*Corresponding Author.

39th Conference on Neural Information Processing Systems (NeurIPS 2025) Track on Datasets and Benchmarks.

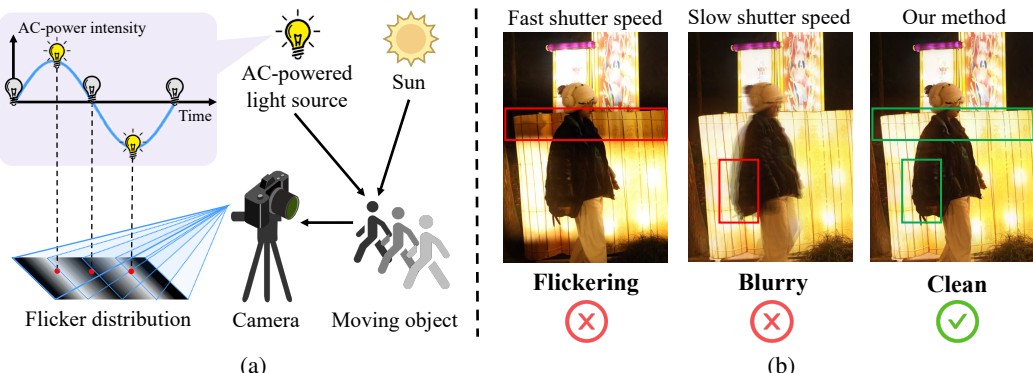

Figure 1: (a) Illustration of flicker formation. The moving object is illuminated by stable light and AC-powered flickering sources. The intensity of the flickering component changes over time (purple area), and each row is exposed at a slightly different moment, leading to a non-uniform brightness distribution across the captured image. (b) Capturing short-exposure images under artificial lighting often results in flicker degradation (red box in the left). Although increasing the exposure time can mitigate flicker artifacts, it introduces motion blur [17, 18, 19] (red box in the middle). Our method effectively removes flicker while preserving fine image details (green box in the right).

digital cameras utilize rolling shutter and line-scan exposure mechanisms [1, 15, 16], in which different rows of the image sensor are exposed at slightly different times. This temporal offset introduces spatial inconsistencies in illumination. Flicker artifacts degrade visual quality, potentially ruining valuable moments for users, especially in scenes with abundant artificial lighting, such as amusement parks, lantern festivals, and cinemas. In addition, flicker impairs the performance of downstream computer vision tasks such as detection, tracking, and recognition [20, 21, 22].

Traditional approaches for flicker mitigation primarily rely on sensor-level solutions, such as integrating flicker detection circuits into camera hardware [23]. Upon detecting flicker, these systems often extend the exposure time to mitigate its effects. However, this introduces motion blur [24, 25, 26, 27, 28], as presented in Figure 1(b). Owing to the adaptability and scalability of deep learning [29, 30], data-driven methods have become the dominant paradigm in image restoration tasks [31, 32, 33]. Their effectiveness heavily depends on the diversity and realism of training datasets. Wong *et al.* [1] propose simulating flicker by superimposing sinusoidal intensity variations onto static images to emulate periodic illumination fluctuations. However, their synthetic data is primarily intended for geo-tagging tasks, rather than for flicker removal. Besides, unlike globally consistent dimness in low-light scenarios [34], flicker artifacts are spatially localized and temporally dynamic. Therefore, single-image flicker removal (SIFR) methods [2, 35] struggle to differentiate flicker from similar dark regions (e.g., shadows), and cannot recover the severely degraded areas due to the lack of pixel context [36]. The intrinsic ambiguity of spatial-only observations results in unreliable restoration.

Modern handheld devices typically capture multiple frames in a single shot, inherently providing rich temporal cues and inter-frame correlations [37]. These cues are highly beneficial for accurate flicker localization and removal, effectively reducing the reliance on explicit priors such as masks in deshadowing [38, 39]. Leveraging this property, multi-frame flicker removal (MFFR) emerges as a more promising and practical solution, particularly under dynamic conditions. However, building a large-scale and high-quality MFFR dataset for dynamic scenes remains highly challenging. On the one hand, capturing a large number of real-world flickering image pairs is labor-intensive and time-consuming [40]. Even with sufficient manpower, the quantity and diversity of flickering patterns that can be collected through real-world capture remain limited. On the other hand, the non-repeatable nature of dynamic scenes makes it nearly impossible to acquire aligned flickering and flicker-free pairs with motion. The lack of such dynamic paired data leads to a critical issue: models tend to misinterpret motion-induced pixel variations as flicker artifacts.

To address these challenges, we construct a comprehensive dataset from three complementary perspectives. First, we propose a flicker synthesis method grounded in Retinex theory [41], capable of generating an unlimited number of flickering images across different scenes. The method explicitly models the interplay between ambient lighting and flickering light sources and supports diverse flicker

patterns caused by common light sources with different rectification modes. Second, to bridge the domain gap between synthetic and real-world datasets, we collect real-world flickering sequences from various static scenes. These sequences serve as a foundation for understanding the spatial and temporal characteristics of real flicker artifacts. Finally, to overcome the inherent non-repeatability of dynamic scenes, we introduce a novel green-screen compositing method. Specifically, we extract foreground subjects with motion from green-screen footage and composite them onto the previously captured flickering backgrounds. This results in a set of flickering image pairs that preserve real flicker degradation while introducing realistic motion dynamics. By integrating these three complementary data sources, we introduce the first MFFR dataset, named *BurstDeflicker*. It comprises an unlimited number of synthetic images, 4,000 real-world flickering image pairs across diverse scenes, and 3,690 green-screen dynamic image pairs generated from the real images.

Our main contributions are summarized as follows: (1) We present BurstDeflicker, the first dataset for multi-frame flicker removal (MFFR), consisting of synthetic, real-world captured, and manually constructed dynamic data. (2) We propose a Retinex-based flicker synthesis method that jointly models ambient and flickering illumination. This enables the scalable generation of synthetic images with diverse and realistic flicker patterns. To overcome the difficulty of acquiring dynamic flickering image pairs, we introduce a green-screen compositing method, which helps mitigate motion ghosting artifacts in multi-frame restoration. (3) Extensive quantitative and qualitative experiments validate the effectiveness of our proposed MFFR dataset. We believe that it will provide a strong foundation for future research in flicker removal.

## 2  Related work

**Hardware-based methods.** Early efforts in flicker removal commonly rely on specialized hardware components, such as photodiodes, to monitor periodic brightness fluctuations in the environment. These systems dynamically adjust camera exposure settings in response, thereby reducing the visibility of flicker artifacts [42]. For instance, Park *et al.* [43] proposed a basic flicker detection and avoidance strategy using a lookup table combined with PID control to modulate exposure timing. Poplin [23] introduced an automatic flicker detection approach optimized for embedded camera systems, aiming for real-time deployment. More recently, neuromorphic vision sensors [44, 45, 46] have opened up new avenues in flicker detection by capturing event-based brightness changes, offering significantly higher temporal resolution than traditional frame-based acquisition. For example, Wang *et al.* [47] proposed a linear comb filter that effectively exploits the high temporal resolution of event-based sensors for flicker removal. Despite their technical advantages, the high cost, limited accessibility, and complex calibration requirements of these methods hinder deployment in consumer-level applications.

**Single image flicker removal.** The term "flicker" is sometimes used to describe brightness discontinuities across video frames [48, 49], such as those found in old films [50]. Restoring such flicker in videos, typically referred to as photometric stabilization [51], is a distinct task from ours. In this paper, we focus on image degradation caused by flicker of AC-powered lights. Several methods [52, 53, 54] have been proposed to suppress flicker, assuming prior knowledge of the lighting conditions. However, the unavailability of such prior information in most real-world scenarios significantly limits their practicality. Yoo *et al.* [35] presented a wide dynamic range system for flicker removal, leveraging long-exposure frames to recover flicker degradation in short-exposure frames. Kim *et al.* [55] introduced a multiplicative model that estimates spatial illumination variations from uniform reflectance areas, which experiences a significant drop in performance when dealing with complex backgrounds. More recently, Lin *et al.* [2] have introduced DeflickerCycleGAN, the first learning-based approach for single-image flicker removal (SIFR). By designing tailored loss functions, Flicker Loss and Gradient Loss, they effectively harness the translation capabilities of CycleGAN [56] to suppress flicker artifacts.

**Flicker removal dataset.** The success of learning-based flicker removal models is heavily dependent on the availability of high-quality paired datasets [57, 58, 59]. However, acquiring a large-scale dataset consisting of both flicker-corrupted and flicker-free image pairs is challenging and labor-intensive, especially for dynamic scenes. Wong *et al.* [1] proposed a flicker synthesis method based on electric network frequency and the rolling shutter mechanism of cameras, primarily for geo-tagging purposes. Then, Lin *et al.* [2] directly used it to construct a flicker removal dataset. As this synthesis approach is originally intended for the geo-tagging task rather than flicker removal, the resulting flickering images lack sufficient variability and realism. Moreover, these synthetic pairs model

flicker removal as the complete elimination of flicker-induced illumination, whereas in reality, the illumination should be adjusted to its effective value rather than entirely removed. As a result, models trained on such data struggle to generalize well to real-world flicker removal tasks.

To address these limitations, we propose a Retinex-based flicker synthesis method tailored for flicker removal. Our method models the interaction between flickering and stable light sources and covers diverse flicker patterns and intensities, better reflecting the variability found in real-world scenarios. We also collect many real-world flickering images and employ a green-screen compositing technique to address the lack of dynamic paired data.

# 3 BurstDeflicker

To the best of our knowledge, there is no publicly available dataset specifically designed for flicker removal, which presents a major obstacle for training and evaluating deep-learning models. To tackle this issue, we present the Burst-Deflicker dataset, which is composed of three subsets: synthetic data, static data captured from the real world, and dynamic green-screen data derived from real static data. These three subsets respectively address the challenges of acquiring large-scale, realistic, and dynamic data, as illustrated in Figure 2. We believe this dataset will

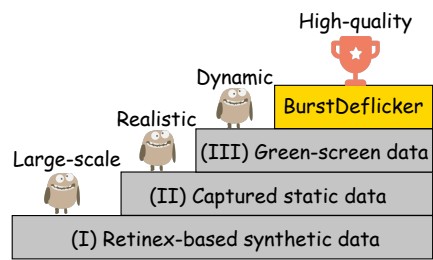

Figure 2: A visual illustration of the three-stage growth of the BurstDeflicker dataset.

serve as a valuable benchmark for the multi-frame flicker removal (MFFR) task and foster further research in this underexplored yet practically important domain.

## 3.1 Retinex-based synthetic flicker

The previous flicker synthesis theory [2] treats the brightness caused by flickering light as harmful. The purpose of this method is to remove the brightness changes caused by flickering illumination. We argue that the flickering illumination should not be completely removed. Instead, the instantaneous flicker illumination should be adjusted to an effective value.

The light in a scene can be categorized into two components: flickering light and ambient light [35], as shown in Figure 1(a). According to the Retinex theory [41], the flickering image $I_{flicker} \in \mathbb{R}^{H \times W \times 3}$ can be expressed as:

$$I_{flicker} = R \odot (L_a + L_f) \tag{1}$$

where $\odot$ is the element-wise multiplication. $R \in \mathbb{R}^{H \times W \times 3}$ represents the reflectance, and $L_a \in \mathbb{R}^{H \times W}$ and $L_f \in \mathbb{R}^{H \times W}$ are the illumination maps of the ambient light and the flickering light, respectively.

The goal of flicker removal is to obtain a flicker-free image $I_{clean} \in \mathbb{R}^{H \times W \times 3}$. In contrast to previous work [2], which model $I_{clean} = R \odot L_a$, we propose that the correct formulation should be $I_{clean} = R \odot (L_a + \overline{L_f})$, where $\overline{L_f}$ denotes the effective value of the flickering light illumination. To derive the relationship between the flickering and flicker-free images, we rewrite Equation (1) as follows:

$$I_{flicker} = R \odot (L_a + \overline{L_f} - \overline{L_f} + L_f) \tag{2}$$

We assume that the ambient light intensity is $k$ times the flickering light intensity. The flickering images can be derived:

$$I_{flicker} = (k+1)R \odot \overline{L_f} \cdot \left(1 + \frac{L_f/\overline{L_f} - 1}{k+1}\right) = I_{clean} \cdot \left(1 + \frac{L_f/\overline{L_f} - 1}{k+1}\right) \tag{3}$$

By changing the background, flicker modes, and the ratio of ambient light to flickering light, we can synthesize flickering images with diverse patterns.

**Light source rectification modes.** In the real world, different light sources have different rectification modes, including full-wave rectified fluorescent lights, half-wave rectified incandescent bulbs, and PWM-rectified LED lights [60, 61]. The sinusoidal alternating current, after undergoing different

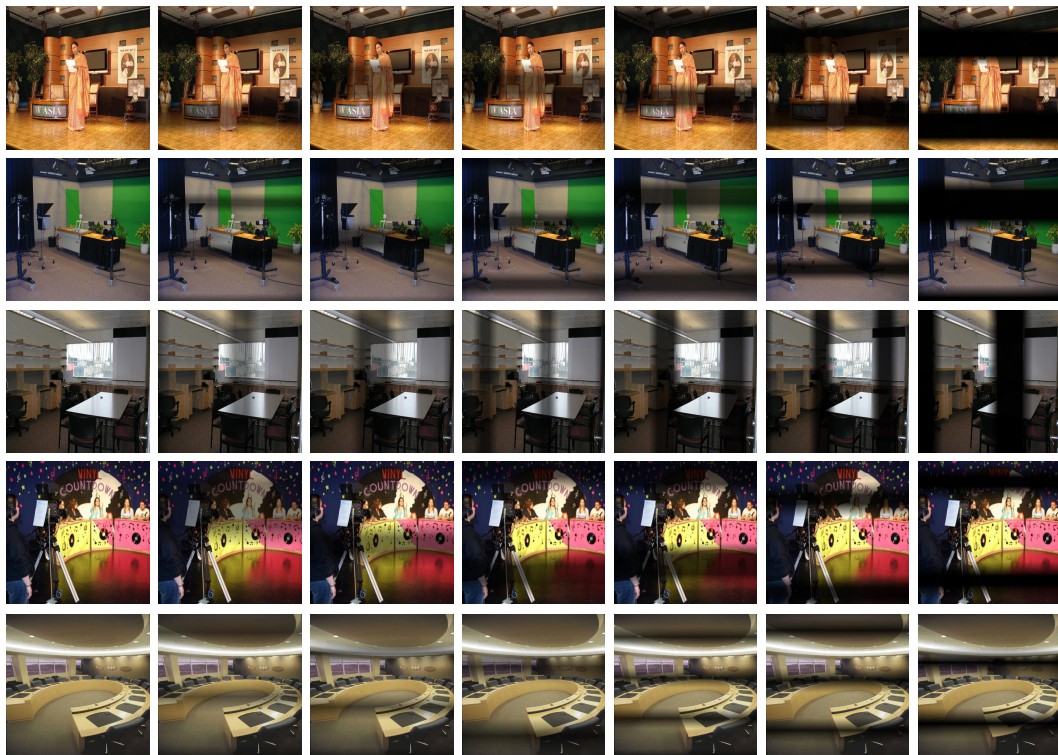

Figure 3: Flicker synthesis results based on the proposed Retinex-based method. Background images are sourced from the indoorCVPR dataset [63]. A pre-training is conducted using the synthetic data, providing a strong initialization for subsequent training on real data.

rectification methods, results in distinct patterns. The flicker caused by light with different rectification modes can be denoted as:

$$
\begin{aligned}
L_{f\sim full} &= A^c \left| \cos\left( 2\pi f_{\text{enf}} \frac{y}{f_{\text{row}}} + \varphi \right) \right| \\
L_{f\sim half} &= A^c \max\left( 0, \cos\left( 2\pi f_{\text{enf}} \frac{y}{f_{\text{row}}} + \varphi \right) \right) \\
L_{f\sim pwm} &= \begin{cases} A^c, & \text{if } \cos\left( 2\pi f_{\text{enf}} \frac{y}{f_{\text{row}}} + \varphi \right) > \cos(\pi D) \\ 0, & \text{otherwise} \end{cases}
\end{aligned}
\tag{4}
$$

where $L_{f\sim full}$, $L_{f\sim half}$ and $L_{f\sim pwm}$ represent the flicker intensity distributions under full-wave, half-wave, and PWM rectification modes, respectively. $f_{enf}$ denotes the electric network frequency and $f_{row}$ represents the row scanning frequency of the camera. $y$ represents the $y$-th row of pixel points along the scanning direction of the sensor. $\varphi$ is the initial phase when capturing images. $D$ is the duty cycle of the PWM rectification mode, with a range of values from 0 to 1. $A^c$ represents the intensity of each RGB channel, which depends on the spectra of the light source [62]. Specific solutions for $L_f$ in Equation 3 including full-wave rectification, half-wave rectification, and PWM, which are derived in Equation 4 using $L_{f\sim full}$, $L_{f\sim half}$, and $L_{f\sim pwd}$, respectively.

Following Lin *et al.* [2], we select the images from the indoorCVPR dataset [63] as background images. We synthesize flicker with the same pattern on each burst sequence, only changing $\varphi$ of the AC power. The range of the intensity ratio $k$ between ambient light and flicker light is set from 0 to 1. According to [1], $f_{enf}$ is 50 or 60 Hz. We use the same $f_{row}$ between 100 kHz and 160 kHz as in [2] for 512×512 resolution. Representative visualizations of the synthetic data are provided in Figure 3.

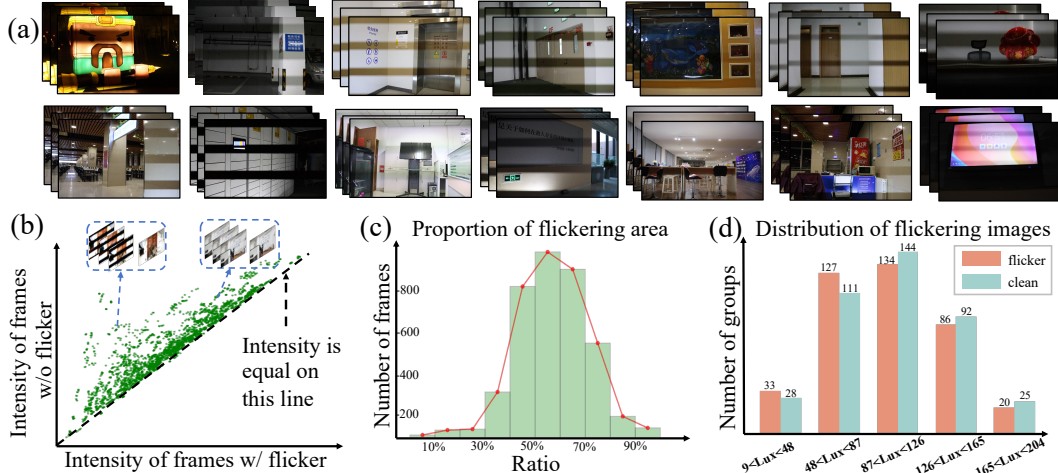

Figure 4: Illustration of the real captured dataset. (a) Example images from our dataset, which include a diverse range of common artificial lighting scenarios. (b) The intensity distributions of flicker and non-flicker frames. (c) The area ratio of flicker degradation per image. (c) The luminance distribution of flickering and clean images across different scenes.

## 3.2 Collection of real-world flickering image pairs

To construct a high-quality dataset containing real-world flickering images, we design a data acquisition pipeline that ensures spatial alignment, high resolution, and scene diversity. A key challenge lies in accurately capturing flicker artifacts while minimizing misalignment between flickering frames and their corresponding ground truth (GT) references.

To address this, we employ a Canon EOS R7 with a 15–150mm f/2.8 lens and a Canon EOS R6 Mark II paired with a 24–105mm f/4–7.1 STM lens, both securely mounted on tripods. To minimize vibration, a remote shutter release is used, and the cameras operate in electronic shutter burst mode to eliminate mechanical jitter during high-speed capture. All recordings are made in manual mode to ensure consistent imaging conditions. First, we reduce the shutter speed to 1/1000-1/2000 seconds to capture flicker artifacts in static scenes. These short exposures capture the high-frequency luminance variations caused by artificial light sources. Second, we capture a clean, flicker-free image using a slow shutter speed of 1/50 or 1/60 seconds, depending on the local electric network frequency [1]. The ISO is adjusted accordingly to ensure that the clean long-exposure image receives the same amount of light as flickering images. This long exposure integrates multiple flicker cycles, effectively suppressing temporal luminance variations and producing an ideal illumination reference.

We capture 10 consecutive frames in a single burst sequence, preserving visible flickering artifacts with precise spatial alignment. We collect flickering sequences across 369 different real-world scenes, including indoor (e.g., offices, supermarkets, subway stations) and outdoor (e.g., LED billboards, parking lots) environments. Since all the flickering images in this subset are captured under static scenes, we refer to this set of 4,000 images as BurstDeflicker-S. Its distribution characteristics and representative sequences are illustrated in Figure 4.

## 3.3 Green-screen compositing flickering image

Since dynamic motion in the real world cannot be exactly replicated, real-world flickering image pairs can only be captured in static scenes. Insufficient exposure to dynamic data may lead the model to misinterpret motion-related pixel variations as flicker, thereby introducing ghosts or misaligned artifacts in the restored images. To address this challenge, we draw inspiration from green-screen compositing techniques widely used in the film industry and propose a novel green-screen approach to simulate realistic motion in flickering sequences. We present a schematic illustration of the motion synthesis process, as shown in Figure 5(a). The synthesis phase takes a group of real-world flickering image pairs as background, and overlays green-screen foregrounds selected from the VideoMatte240K dataset [64]. For each scene, we manually select foreground clips that are semantically and spatially

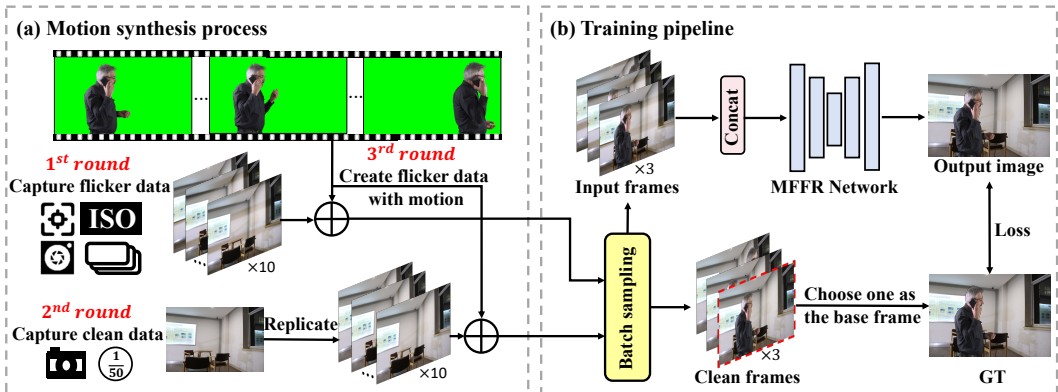

Figure 5: The motion synthesis process and training pipeline. (a) The green-screen footage is selected from the VideoMatte240K dataset [64]. The compositing of green-screen foregrounds with the backgrounds is manually performed using Adobe After Effects. (b) Given a sequence of flickering frames, we select three frames as input to form a training batch, with the target being a single clean reference image. The synthetic dataset and BurstDeflicker-S also follow this pipeline.

compatible with the background, and use the corresponding alpha masks to perform high-quality compositing. Since the clean GT image is a single frame, we replicate it ten times and overlay the same ten foreground clips used in the flickering sequence. This compositing strategy allows us to simulate realistic dynamic content while preserving authentic flicker artifacts in the background, effectively addressing the lack of dynamic scenes in the dataset. Using this approach, we construct a semi-synthetic green-screen dataset consisting of 3,690 images with motion, which we denote as BurstDeflicker-G. It serves as a crucial step toward improving the robustness and generalization of flicker removal models in dynamic, real-world scenarios. More details can be found in the supplementary materials.

We obtain 4,000 real-world static flickering images (referred to as BurstDeflicker-S) along with their corresponding dynamic green-screen image pairs (referred to as BurstDeflicker-G). These datasets are partitioned into training and testing splits using an 8/2 ratio. To simulate natural handheld motion, we introduce synthetic camera shake by applying random rotations in the range of $[-3°, 3°]$ and translations within $[-5, 5]$ pixels to the burst sequences. Besides, to facilitate model training, input images are often cropped into small patches in burst image super resolution [36, 65, 66]. However, flicker artifacts are localized and exhibit periodic patterns along the line-scan direction. To preserve this periodicity, we resize the images during training instead of cropping them.

**Training pipeline.** We present the MFFR training pipeline in Figure 5(b). For each training iteration, we randomly sample three frames at intervals of 1 to 3 to simulate different varying camera capture rates. These three frames are augmented and concatenated, then fed into different MFFR networks for restoration. Note that the final output is a single image, corresponding to one of the three flickering frames. Owing to the significant expense and effort involved in collecting paired MFFR data, training a model from scratch with a large real dataset is impractical [67]. Following the previous burst super-resolution work [36], we use the proposed Retinex-based synthesis method to generate a large dataset for pre-training the network. The pre-trained model acts as a strong initialization, then undergoes fine-tuning on our BurstDeflicker dataset for real-world flicker removal.

## 4 Experiments

### 4.1 Comparison with previous work

**Experimental settings.** Low-light enhancement is similar to the flicker removal task, so we use the pre-trained model of the representative Retinexformer [68] for flicker removal. We also test the performance of the state-of-the-art SIFR method, DeflickerCycleGAN, proposed by Lin *et al.* [2]. We conduct a benchmark using three baselines trained on our dataset: Burstormer [69] for burst image restoration, HDRTransformer [70] for HDR imaging, and Restormer [71] for single-image restoration. Specifically, we change Restormer's input channels from 3 to 9 and concatenate the input

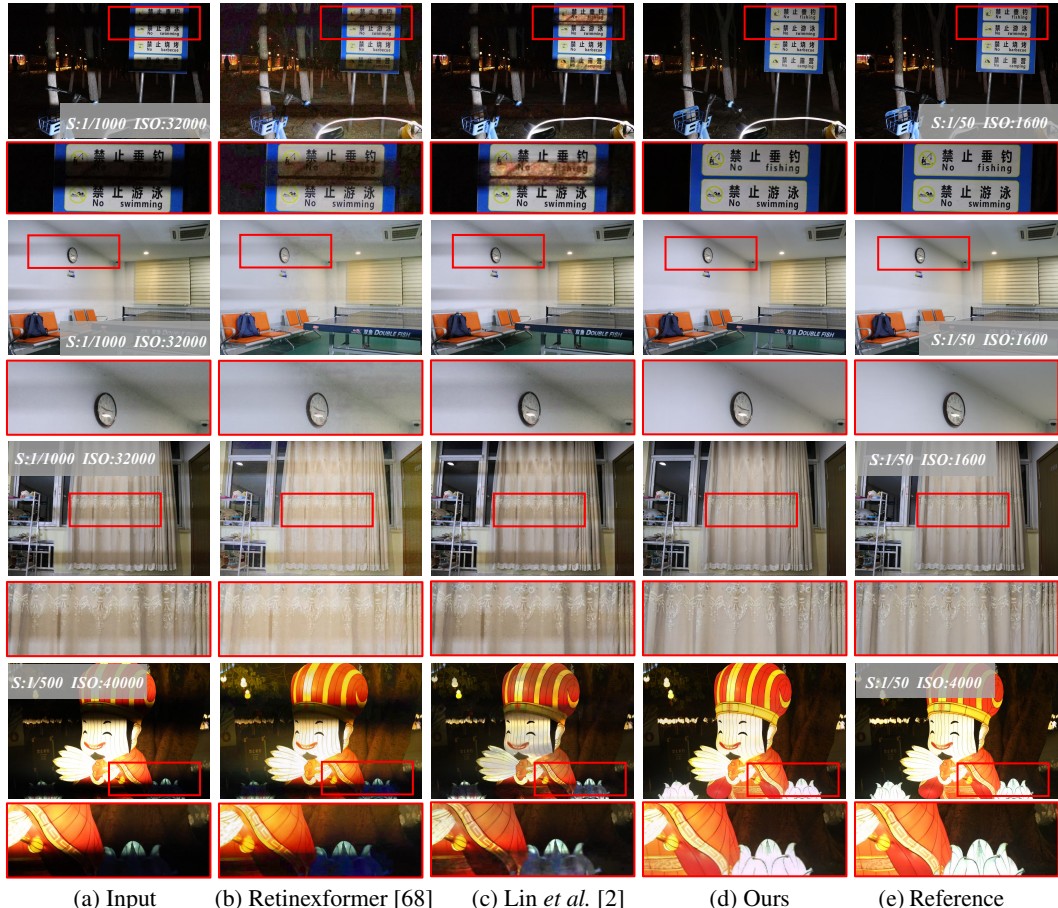

| (a) Input | (b) Retinexformer [68] | (c) Lin *et al.* [2] | (d) Ours | (e) Reference |

Figure 6: Visual results of flicker removal on the static test data. Our results are obtained by training the Restormer on our dataset. It demonstrates the best performance across diverse flickering scenes.

frames along the channel dimension to achieve multi-frame fusion. Due to the memory limitation (24 GB) of the RTX 3090, we reduce the parameters of Restormer [71]. Specifically, we reduce the number of refinement blocks in Restormer [71] from 4 to 2, and set the feature channel dimension to 32 instead of the default 48. Besides, since Burstformer [69] originally takes 8-frame inputs while Restormer [71] is designed for single-frame inputs, we modify both models to take 3-frame inputs for consistency. More experimental details, settings, and qualitative comparisons are presented in the supplementary materials.

**Qualitative comparison.** We show the visual comparison of flicker removal on the test data of BurstDeflicker-S in Figure 6. The low-light image enhancement method, Retinexformer [68], globally brightens the image and introduces color shifts, which is undesirable for the goal of flicker removal. The SIFR method of Lin *et al.* [2] has little effect on flicker removal, especially in cases of severe flicker and insufficient lighting. Restormer, trained on our dataset, demonstrates effective flicker removal in dealing with mild indoor flicker and strong nighttime flicker.

**Quantitative comparison.** We evaluate the performance of different methods on the static test set using three full-reference metrics, including PSNR, SSIM [72], and LPIPS [73]. Since the test set and training set come from the same domain, the overfitted model may achieve better results. Therefore, we capture 50 dynamic test sequences using various mobile devices and consumer cameras to evaluate the model's performance and robustness in real-world dynamic scenarios. Since the ground truths of real dynamic images are unavailable, we employ MUSIQ [74], BRISQUE [75], and PIQE [76] as no-reference evaluation metrics.

The flicker removal results of different methods are shown in Table 1. Retinexformer performs poorly in flicker removal, indicating that training on a low-light dataset alone is insufficient to

Table 1: Quantitative results of different methods. Note that '*' indicates models that are not trained on our dataset but directly tested using their original versions. To ensure a fair comparison of dataset effectiveness, we retrain Retinexformer [68] and the model of Lin *et al.* [2] on our dataset using single-frame input, following the same setup as in their original work.

| Method | Flops (G) | Params (M) | Static test data | | | Dynamic test data | | |
|---|---|---|---|---|---|---|---|---|
| | | | PSNR ↑ | SSIM ↑ | LPIPS ↓ | MUSIQ ↑ | PIQE ↓ | BRISQUE ↓ |
| *Retinexformer [68] | 69.23 | 1.61 | 15.704 | 0.707 | 0.213 | 53.596 | 50.269 | 30.242 |
| *Lin *et al.* [2] | 509.80 | 92.08 | 20.358 | 0.838 | 0.134 | 55.228 | 43.875 | 25.121 |
| Lin *et al.* [2] | 509.80 | 92.08 | 26.408 | 0.875 | 0.102 | 58.131 | 35.710 | 22.102 |
| Retinexformer [68] | 69.23 | 1.61 | 27.212 | 0.885 | 0.081 | 58.249 | 35.942 | 21.648 |
| Burstormer [69] | 141.05 | 0.17 | 29.439 | 0.910 | 0.056 | 58.527 | 37.014 | 20.451 |
| HDRTransformer [70] | 272.12 | 1.04 | 30.031 | 0.914 | 0.054 | 59.069 | 37.292 | 21.588 |
| Restormer [71] | 149.01 | 7.92 | **30.634** | **0.918** | **0.045** | **59.097** | **34.896** | **19.324** |

Table 2: Ablation study of Restormer trained on different parts of the dataset. The synthetic dataset enhances the model's robustness and overall performance. In particular, the green-screen data (BurstDeflicker-G) significantly improves the model's effectiveness in real-world dynamic scenarios.

| Training data | | | Static test data | | | Dynamic test data | | |
|---|---|---|---|---|---|---|---|---|
| Synthetic data | BurstDeflicker-S | BurstDeflicker-G | PSNR ↑ | SSIM ↑ | LPIPS ↓ | MUSIQ ↑ | PIQE ↓ | BRISQUE ↓ |
| ✓ | | | 24.483 | 0.862 | 0.122 | 57.096 | 39.726 | 23.755 |
| | ✓ | ✓ | 30.481 | 0.915 | 0.053 | 58.011 | 37.498 | 21.523 |
| ✓ | ✓ | | **30.645** | 0.916 | 0.052 | 58.431 | 36.259 | 20.338 |
| ✓ | ✓ | ✓ | 30.634 | **0.918** | **0.045** | **59.097** | **34.896** | **19.324** |

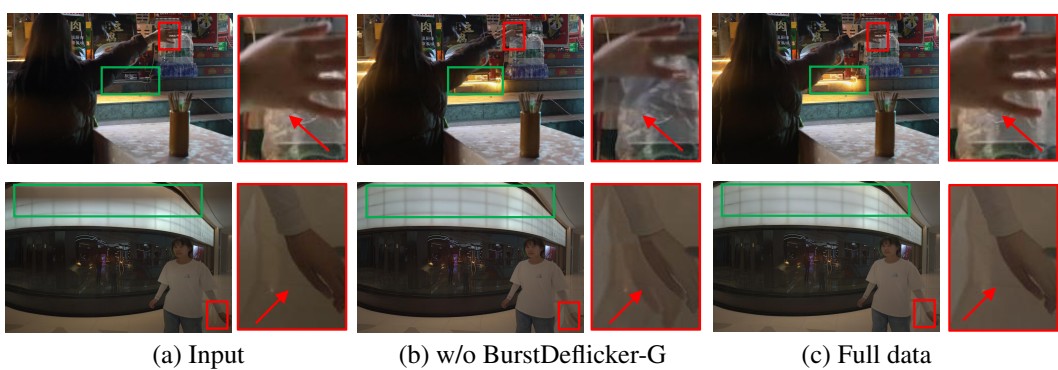

(a) Input      (b) w/o BurstDeflicker-G      (c) Full data

Figure 7: The visual comparison on the dynamic test data of Restormer trained with/without BurstDeflicker-G. The introduction of BurstDeflicker-G helps reduce motion ghosting artifacts (red boxes) without compromising the flicker removal performance (green boxes).

address the flicker problem. We re-train Retinexformer [68] and Lin [2]'s network on our dataset, achieving significant improvements, which demonstrates the effectiveness of the BurstDeflicker dataset. Additionally, we train three representative networks using a burst size of three, all of which demonstrate strong flicker removal performance, with Restormer achieving the best results.

### 4.2 Ablation study

**Parts of BurstDeflicker.** Synthetic data helps reduce the risk of overfitting and enhances the model's robustness, as demonstrated in the second and fourth rows of Table 2. The BurstDeflicker-G subset helps improve the model's performance on the dynamic test set, as shown in the third and fourth rows of Table 2. To intuitively explain the improvement caused by green-screen data, we present a visual comparison of flicker removal on the handheld-captured data in Figure 7. The model without green-screen data may introduce motion ghosts. The essence of the green-screen method is to allow the model to "see" flickering image pairs with motion. By training on such data, the model learns to distinguish whether pixel value changes are caused by motion or flicker.

Table 3: The test results with different numbers of burst image inputs. Multi-frame flicker removal achieves better performance compared to single-image restoration. Due to the redundancy between adjacent frames, the performance gain from increasing the input from two to three frames is smaller than that from one to two frames, which indicates the marginal effect of adding more input frames.

| Input | Flops (G) | Static test data | | | Dynamic test data | | |
|---|---|---|---|---|---|---|---|
| | | PSNR ↑ | SSIM ↑ | LPIPS ↓ | MUSIQ ↑ | PIQE ↓ | BRISQUE ↓ |
| Single image | 148.55 | 27.310 (+0.000) | 0.891 (+0.000) | 0.069 (-0.000) | 58.664 | 35.821 | 20.010 |
| Burst-2 | 148.78 | 30.264 (+2.594) | 0.915 (+0.024) | 0.048 (-0.021) | 58.786 | 35.277 | 19.798 |
| Burst-3 | 149.01 | 30.634 (+3.324) | 0.918 (+0.027) | 0.045 (-0.024) | **59.097** | **34.896** | **19.324** |

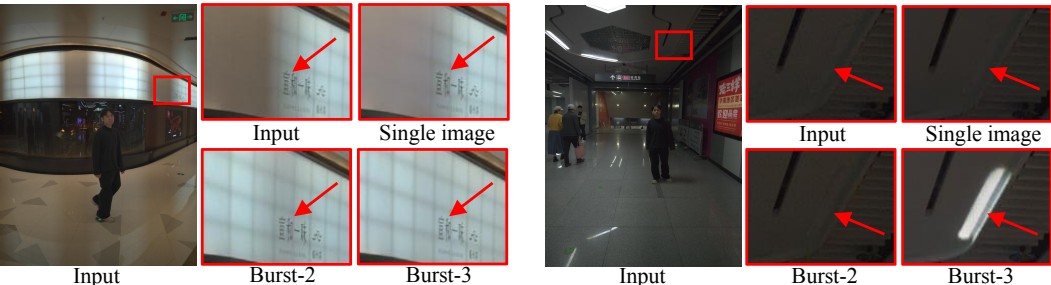

Figure 8: The visual comparison of flicker removal using different numbers of input frames. As the number of input frames increases, the model exhibits progressively better restoration performance for banding artifacts (left). Moreover, when the number of input frames reaches three, the model successfully restores the originally extinguished light source (right).

**Number of input frames.** Following previous multi-frame restoration tasks [36, 57], we train Restormer with varying numbers of input frames to validate the effectiveness of the MFFR strategy. We present the performance of Restormer [71] under varying numbers of input frames, as shown in Table 3. The PSNR improvement from 2 to 3 input frames (0.730 dB) is less significant than that from 1 to 2 input frames (2.594 dB), which can be attributed to the overlapping clean regions among multiple input frames. We provide a visual comparison of restoration results using different numbers of input frames, as depicted in Figure 8.

## 5 Conclusion

In this paper, we introduced the first multi-frame flicker removal (MFFR) dataset, BurstDeflicker, which consists of large-scale synthetic images, 4,000 real-world captured image pairs, and 3,690 manually created image pairs with motion. The synthetic method simulated the interaction between ambient light and flicker light, considering various flicker patterns for pretraining the flicker removal model. The real-world captured dataset was used to fine-tune the model for improved performance. Furthermore, since paired motion images are difficult to capture, we proposed a motion embedding method based on the green-screen technique, which helped mitigate motion ghosting issues in multi-frame fusion. Comprehensive experiments demonstrated the effectiveness of our MFFR method and dataset, which can facilitate future research in flicker removal.

**Limitation.** When an AC-powered light source serves as the sole illumination source, the resulting flicker degradation can be particularly severe, potentially leading to noticeable color shifts in the restoration results, as illustrated in the fourth row of Figure 6. Although MFFR methods can leverage multi-frame information for more accurate restoration, they still struggle when the available frames lack complete scene content. Besides, when there is significant misalignment between frames (e.g., strong handheld jitter), the performance of multi-frame restoration may degrade to that of single-frame methods. To address these challenges, future work could focus on designing more efficient network architectures that are specifically tailored to the unique features and priors of flicker.

**Acknowledgement.** This work was supported by Shenzhen Science and Technology Program (No. JCYJ20240813114229039), National Natural Science Foundation of China (No. 624B2072), Natural Science Foundation of Tianjin (No.24JCZXJC00040), Supercomputing Center of Nankai University, and OPPO Research Fund.

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
