# OpenReview forum: "BurstDeflicker: A Benchmark Dataset for Flicker Removal in Dynamic Scenes"
_NeurIPS.cc/2025/Datasets_and_Benchmarks_Track — NeurIPS 2025 Datasets and Benchmarks Track poster_

### Official Review · Reviewer_sgXN · 2025-06-14

**Rating:** 4
**Confidence:** 4

**Summary:**

This paper proposes BurstDeflicker, the first multi-frame flicker removal (MFFR) dataset, using Retinex-based synthetic data, static real images, and dynamic green-screen data to solve dynamic-scene flicker data scarcity and enhance real-world generalization.

**Dataset Code Accessibility:**

Yes

**Ethical Comments:**

No significant ethical concerns exist.

**Ethical Considerations:**

No, there are no or only very minor ethics concerns

**Final Justification:**

I appreciate the time you took to respond and clarify. I will maintain my score.

**Limitations Weaknesses:**

**Weaknesses**

1. The authors fail to quantify the domain gap between green-screen compositing and real-world lighting interactions (e.g., inconsistent shadows/reflections on moving subjects), risking model overfitting to synthetic artifacts.

2. Dynamic scene evaluation relies solely on no-reference metrics (MUSIQ/PIQE/BRISQUE) without human subjective assessment (e.g., MOS), despite flicker being a highly perception-sensitive artifact.

3. Static scenes lack metadata on illumination distribution (low-light vs. harsh-light ratios), while dynamic data exclusively uses VideoMatte240K's human-centric foregrounds without testing non-human motions (e.g., vehicles).

4. The method trains models exclusively on 3-frame bursts without validating performance on longer sequences (e.g., 10+ frames common in mobile photography), potentially limiting real-world applicability where temporal redundancy could enhance restoration.

**Strengths Contributions:**

**Strength**

1. The work proposes a novel multi-frame flicker removal dataset covering synthetic, static, and dynamic scenes.

2. The authors develop a physics-based synthesis method modeling ambient-flicker light interactions and rectification modes.

3. The solution effectively addresses dynamic scene non-repeatability via green-screen compositing for real-world motion generalization.

---

> ### Author Rebuttal · Authors · 2025-07-30
>
> We sincerely appreciate your valuable comments on our work. We have addressed your concerns as follows.
>
> > *The authors fail to quantify the domain gap between green-screen compositing and real-world lighting interactions (e.g., inconsistent shadows/reflections on moving subjects), risking model overfitting to synthetic artifacts.*
>
> According to the quantitative (Tables 1 and 2 in manuscript) and qualitative results (Figures 5 and 6), the risk of overfitting the model on synthetic artifacts is almost non-existent due to the availability of real datasets.
> The green-screen data demonstrates greater benefits than its shortcomings, primarily in suppressing motion ghosting artifacts, which exert a more substantial impact in real scenarios as demonstrated in the ablation study (Table 2 and Figure 6).
>
> Acquiring authentic dynamic paired flicker datasets remains highly challenging and will constitute a key research direction for future work.
>
> > *Dynamic scene evaluation relies solely on no-reference metrics (MUSIQ/PIQE/BRISQUE) without human subjective assessment (e.g., MOS), despite flicker being a highly perception-sensitive artifact.*
>
> Thank you for your valuable advice. In response, we conducted a user study involving 10 participants. Each participant was asked to score all restored images (1-5 score), where the original input image was presented as a reference for score 1, and the ground truth (GT) image as a reference for score 5.
>
> |       | Lin et al. | Retinexformer | Burstormer | HDRTransformer | Restormer |
> | ----- |:----------:|:-------------:|:----------:|:--------------:|:---------:|
> | Score | 2.71       | 3.54          | 3.75       | 3.92           | 4.52      |
>
> > *Static scenes lack metadata on illumination distribution (low-light vs. harsh-light ratios), while dynamic data exclusively uses VideoMatte240K's humancentric foregrounds without testing non-human motions (e.g., vehicles).*
>
> **Illumination distribution**. We used the average brightness of each image as the threshold to differentiate low-light and harsh-light regions. The statistics of all flicker images (harsh-light ratios) are summarized as follows:
>
> |        | (10%,20%] | (20%,30%] | (30%,40%] | (40%,50%] | (50%,60%] | (60%,70%] | (70%,80%] |
> | ------ |:---------:|:---------:|:---------:|:---------:|:---------:|:---------:|:---------:|
> | number | 11        | 147       | 687       | 1259      | 1066      | 335       | 20        |
>
> **Motion footage**. Before creating the green-screen dataset, we collected flicker scenes encountered by 104 users, and all of these scenes occurred in human-active regions.
> Therefore, considering that most flicker scenarios involve human-related activities, we primarily choose human motion synthesis as the benchmark.
>
> If there is a need to model other motion scenarios, our paradigm can be easily extended by incorporating other motion materials.
>
> In response to the reviewer's concerns regarding non-human movements, we evaluated our model on several scenes featuring non-human motion, including moving suitcases, office chairs, computers, and cars in underground parking lots. BurstFlicker-G is still effective in suppressing the ghosting artifacts produced in these scenarios, as confirmed by all 10 users who participated in the evaluation.
>
> > *The method trains models exclusively on 3-frame bursts without validating performance on longer sequences (e.g., 10+ frames common in mobile photography), potentially limiting real-world applicability where temporal redundancy could enhance restoration.*
>
> Thank you for your professional advice and consideration of practical implementation. Actually, we selected three frames based on two practical considerations:
>
> (1) In mobile HDR imaging, around 10 frames of images are indeed captured for synthesizing the final image. However, only 3-4 of them are short-exposure images which has flicker. Therefore, although we can use more burst frames to train the model, the model cannot access the same number of short exposure frames during inference.
>
> (2) Choosing more frames may result in a wider range of motion, increase the likelihood of ghosting.
>
> Besides, each group of our dataset comprises 10 consecutive images. If necessary, the followers can design a model with more frames for training, and our dataset can also support it. For example, we extended the ablation study of Restormer with different frames (Table 3 in the manuscript) as shown below.
> We observed that although increasing the number of frames initially improves performance, the gain gradually saturates and eventually starts to decline.
> Moreover, visualizations on the Burst-10 setting reveal more severe motion ghosting artifacts.
>
> |          | Flops(G) | PSNR $\uparrow$ | SSIM $\uparrow$ | LPIPS $\downarrow$ | MUSIQ $\uparrow$ | PIQE $\downarrow$ | BRISQUE $\downarrow$ |
> | -------- |:--------:|:---------------:|:---------------:|:------------------:|:----------------:|:-----------------:|:--------------------:|
> | Burst-1  | 148.55   | 27.310          | 0.891           | 0.069              | 58.664           | 35.821            | 20.010               |
> | Burst-2  | 148.78   | 30.264          | 0.915           | 0.048              | 58.786           | 35.277            | 19.798               |
> | Burst-3  | 149.01   | 30.634          | 0.918           | **0.045**          | **59.097**       | **34.896**        | 19.324               |
> | Burst-5  | 149.46   | 30.652          | **0.920**       | 0.047              | 59.043           | 34.928            | **19.211**           |
> | Burst-10 | 150.59   | **30.670**      | 0.915           | 0.050              | 58.921           | 35.165            | 19.461               |
>
> Thank you again for your valuable suggestions. These suggestions have greatly helped us improve the analysis of the dataset and make this work more comprehensive.

---

> > ### Comment · Reviewer_sgXN · 2025-08-05
> > **Response to Authors**
> >
> > I appreciate the time you took to respond and clarify. I will maintain my score.

---

> > > ### Author Response · Authors · 2025-08-05
> > >
> > > Dear Reviewer sgXN,
> > >
> > > Thank you again for your feedback and insightful suggestions.
> > > We will be sure to incorporate the experiments and details you have
> > > recommended into the revised version of our paper.
> > >
> > > Sincerely,
> > >
> > > Authors of paper 862

---

### Official Review · Reviewer_euf9 · 2025-06-19

**Rating:** 4
**Confidence:** 4

**Summary:**

In this submission, the authors propose BurstDeflicker, a data collection composed of simulated flickering effects and up to 4000 real world images showcasing flickering behavior characterizing light sources under alternative current (AC) power supply. Naturally, they propose a image burst-based training pipeline for general image restoration algorithms that provides sufficient information for lost details restoration through multiple scene observations. The pipeline shows promising potential with both quantitative and qualitative evaluations proving the enhanced performance of current literature deep neural networks, through the proposed training framework.

**Additional Feedback:**

A rather positive feedback is possible  based on the need to develop the task, but in terms of identifying the strategies best to apply in the studied task, extensive evaluation is still needed, and if properly done, likely to drive an extensive rewriting for Sections 4 and 5. The image overlay proposed in BurstFlicker-G might need extensive re-thinking.

**Dataset Code Accessibility:**

Yes

**Dataset Code Comments:**

The data is accessible on Kaggle, and the data structure is easy to navigate, especially if starting from the submitted loaders.

**Ethical Considerations:**

No, there are no or only very minor ethics concerns

**Final Justification:**

Most of the concerns were addressed by the authors in their rebuttal. Additionally, comparisons including current image restoration state-of-the-art models were provided in the rebuttal, and will hopefully appear in the final version. Therefore an increased final rating was possible.

**Limitations Weaknesses:**

1. The model assumes the behavior of lighting under an AC power supply. Currently, LED lights are becoming increasingly popular, in both highly professional studio setups and outdoors areas (e.g. street advertisement). Is this category of lights represented in the real-data sample? How do different models cope with the faster transition under a Pulse Width Modulation (PWM) signal?
2. At a number of 4000 real images, and a burst of 10 images per scene, the real data likely consists of 400 reference images. This number is rather low, and unlikely to grant data-level representation for a wide category of factors (pulse frequency, light shape, relative size of the captured geometry under the light source influence).
3. Light source shape has significant influence in the behavior of the studied flickering effect. How many types of light sources were considered in real data collection, given that the provided details describe a band flickering model? Is pulse flickering also represented?
4. The model seems to assume a singular light source in the analyzed image. However, in practice, multiple lighting is very common, especially for images showing wider geometries.
5. Did the authors take into effect the possibility for a color (or multi-color) lighting model characterizing single (or multiple) lighting sources?
How do the proposed models behave on such data samples, after being trained in the described framework?
6. The benchmark proposed can be easily extended to different classes of IR algorithms (CNN-based such as NAFNET, MPRNet, HINET,  transformer backbones such as Restormer, Uformer, HAT, frequency-representation models such as SFNet, FourierFormer, IFBlend, state-space models live MambaIR, MambaIRv2). Such comparisons would offer crucial insights in designing models tailored for the de-flickering task.
7. For BurstFlicker-G, the some of the foregrounds added on top of flickering backgrounds do not have a realistic look, and the flickering effect observed for the backgrounds is not consistent with the added foregrounds. The scenario in which the appearence of the foreground is independent from the flickering model applies e.g. in the case where the subject is in front of an LED display, but even in this case some lighting effects would be observed on the subject. This data generation part needs extensive validation and a potential re-design. Some comments from the authors would be great, regarding how flickering was kept consistent in the image overlay, and if the assumption of independent foregrounds/backgrounds really holds. Is the model validated for cases in which flickering affects the image subject up to some degree?

**Strengths Contributions:**

1. The dataset contribution is surely important. The fact that the generated synthetic data is validated against real data, even if acquired under a set of controlled factors, is a definitive plus.
2. For a rather sub-developed image restoration field, including rather actual state-of-the-art methods from the broader restoration field in a benchmark will prove beneficial if aiming for production-ready solutions.
3. Including both solutions based on CNN and Transformer backbones gives some insights regarding the complexity of the dependencies needed to restore the flickering-affected image.
4. The manuscript is well organized, and the level of details provided is sufficient for a good understanding of the method, including its limitations.

---

> ### Author Rebuttal · Authors · 2025-07-30
>
> We sincerely thank you for reviewing our paper patiently and providing us valuable feedback. The raised concerns are addressed as follows.
>
> > *The model assumes the behavior of lighting under an AC power supply. Currently, LED lights are becoming increasingly popular, in both highly professional studio setups and outdoors areas (e.g. street advertisement). Is this category of lights represented in the real-data sample? How do different models cope with the faster transition under a Pulse Width Modulation (PWM) signal?*
>
> LED lights are also driven by alternating current, though they utilize a different modulation scheme (as considered in Equation 4). In real-world scenarios, we intentionally captured PWM-induced flicker images, which account for approximately 32.5% of the dataset (e.g., samples 0001 and 0004 in the dataset).
>
> > *At a number of 4000 real images, and a burst of 10 images per scene, the real data likely consists of 400 reference images. This number is rather low, and unlikely to grant data-level representation for a wide category of factors (pulse frequency, light shape, relative size of the captured geometry under the light source influence).*
>
> For multi-frame/Burst tasks, this order of magnitude is common, such as BurstSR (200 groups), REDS (300 groups) for burst super-resolution, SID (161 groups) and SDSD (150 groups) for burst low-light image enhancement, and NTIRE-HDR (29 groups), Mobile-HDR (251 groups) for HDR. For a more detailed comparison, please refer to Table 1 in the supplementary materials.
> The current scale of the dataset is already sufficient to support model training, and the experimental results demonstrate its effectiveness. Thank you for your valuable suggestions, and we will continue improving our dataset collection, which will better equip us to handle challenging scenarios.
>
> > *Light source shape has significant influence in the behavior of the studied flickering effect. How many types of light sources were considered in real data collection, given that the provided details describe a band flickering model? Is pulse flickering also represented?*
>
> According to our statistics, the shapes of light sources in the dataset can be categorized into four: line, area (e.g., screens, billboards), point, and volumetric (e.g., lanterns, illuminated buildings).
> As for the types of light sources, there are three main categories: full-wave, half-wave, and PWM (refer to line 164 in the main text for details). Among them, pulse-induced flicker (PWM) accounts for 32.5% of the dataset. More detailed statistics are provided in the table below.
>
> |        | Motion types | Light shapes | Indoor: outdoor | Day: night    | Full-wave: half-wave: PWM | AC-power light intensity (low:medium: high) |
> | ------ |:------------:|:------------:|:---------------:|:-------------:|:-------------------------:|:-------------------------------------------:|
> | Number | 29           | 4            | 86.5% : 13.5%   | 44.5% : 55.5% | 20% : 47.5% : 32.5%       | 22% : 43% : 35%                             |
>
> > *The model seems to assume a singular light source in the analyzed image. However, in practice, multiple lighting is very common, especially for images showing wider geometries.*
>
> Although our synthetic model has improved compared to previous methods, it is still unable to synthesize the effect of multi-light source interaction, which will be the future research direction. However, 45% of the captured real-world data contained multi-light sources, which, together with synthetic data, constructed a large-scale and relatively realistic benchmark.
>
> > *Did the authors take into effect the possibility for a color (or multi-color) lighting model characterizing single (or multiple) lighting sources? How do the proposed models behave on such data samples, after being trained in the described framework?*
>
> Yes, we did consider multiple colors. The approach in the synthesis process is similar to our previous work (DeflickerCycleGAN), which involves adjusting the proportions of the three color channels to obtain synthesized flicker effects in different colors.
> Our real-world dataset also includes flicker artifacts captured under various colored lighting conditions, such as green fluorescent lights, warm yellow lights, and colored LED displays.
> We conducted an evaluation on non-white light flicker, and the results are presented below.
>
> | Colored lighting condition    | PSNR  $\uparrow$ | SSIM $\uparrow$ | LPIPS $\downarrow$ |
> | ----------------------------- |:----------------:|:---------------:|:------------------:|
> | Lin et al. (previous dataset) | 20.213           | 0.824           | 0.131              |
> | Ours                          | 30.421           | 0.907           | 0.046              |
>
> > *The benchmark proposed can be easily extended to different classes of IR algorithms (CNN-based such as NAFNET, MPRNet, HINET, transformer backbones such as Restormer, Uformer, HAT, frequency-representation models such as SFNet, FourierFormer, IFBlend, state-space models live MambaIR, MambaIRv2). Such comparisons would offer crucial insights in designing models tailored for the de-flickering task.*
>
> Thank you very much for your constructive suggestion. We have conducted these experiments accordingly. It is worth noting that due to the 24GB memory limitation of the RTX 3090 GPU, we reduced the model sizes of MPRNet, Restormer, MambaIR, and MambaIRv2 during training.
> Specifically, the number of features in MPRNet was set to 24 instead of the original 40. For Restormer, the number of refinement blocks was reduced from 4 to 2, and the feature dimension was decreased from 48 to 32. In MambaIR, the network depth was set to 2 and the feature dimension to 48. For MambaIRv2, the feature dimension was set to 24. We believe that this more comprehensive benchmark can facilitate research in the field of flicker removal.
>
> |               | PSNR $\uparrow$ | SSIM $\uparrow$ | LPIPS $\downarrow$ | MUSIQ $\uparrow$ | PIQE $\downarrow$ | BRISQUE $\downarrow$ |
> | ------------- |:---------------:|:---------------:|:------------------:|:----------------:|:-----------------:|:--------------------:|
> | NAFNET        | 28.271          | 0.881           | 0.062              | 58.538           | 37.343            | 21.513               |
> | SFNet         | 28.451          | 0.879           | 0.060              | 58.573           | 37.131            | 20.492               |
> | MPRNet        | 28.498          | 0.886           | 0.050              | 58.246           | 36.751            | 20.341               |
> | HAT           | 28.915          | 0.887           | 0.072              | 58.417           | 36.721            | 21.294               |
> | FourierFormer | 29.492          | 0.910           | 0.056              | 58.669           | 36.912            | 20.648               |
> | IFBlend       | 29.481          | 0.907           | 0.053              | 58.221           | 36.472            | 20.579               |
> | HINET         | 29.552          | 0.890           | 0.046              | 58.425           | 36.692            | 20.165               |
> | MambaIR       | 30.221          | 0.914           | 0.048              | 58.429           | 35.828            | 19.590               |
> | Uformer       | 30.528          | 0.911           | 0.050              | 58.621           | 35.825            | 19.451               |
> | Restormer     | 30.634          | **0.918**       | **0.045**          | **59.097**       | 34.896            | 19.324               |
> | MambaIRv2     | **30.879**      | 0.915           | 0.048              | 58.781           | **34.621**        | **19.210**           |
>
> > *For BurstFlicker-G, the some of the foregrounds added on top of flickering backgrounds do not have a realistic look, and the flickering effect observed for the backgrounds is not consistent with the added foregrounds. The scenario in which the appearence of the foreground is independent from the flickering model applies e.g. in the case where the subject is in front of an LED display, but even in this case some lighting effects would be observed on the subject. This data generation part needs extensive validation and a potential re-design. Some comments from the authors would be great, regarding how flickering was kept consistent in the image overlay, and if the assumption of independent foregrounds/backgrounds really holds. Is the model validated for cases in which flickering affects the image subject up to some degree?*
>
> In fact, previous synthesis methods treated flicker as a uniform degradation for synthesis, making it impossible to distinguish between foreground and background. This is why we need to capture real flicker datasets. For real static scenes we capture, there is a distinction between the foreground and the background. Training the model on this data enables it to learn the interaction between scene structure and flicker effects.
>
> For BurstFlicker-G, its function is not to distinguish between foreground and background, but to suppress ghosting caused by motion across multiple frames.
> Although green-screen images may be a little unrealistic, the experimental results (Table 2 and Figure 6) indicate that the advantages of incorporating BurstFlicker-G outweigh its limitation.
>
> Regarding "how flickering was kept consistent in the image overlay", we believe that a generative model could potentially learn the distribution characteristics of flicker using our dataset, rather than simply applying uniform flicker through handcrafted formulas.
>
> Finally, flicker inevitably impacts subjects in the dynamic scenes, so obtaining more realistic flicker data with motion remains a research direction in the future.
>
> We sincerely appreciate your valuable comments. We hope our response can address your concerns.

---

> > ### Comment · Reviewer_euf9 · 2025-08-05
> > **Post rebuttal feedback**
> >
> > Many thanks for the many insights provided in the rebuttal. While the statistics provided clear some of the doubts and answer the questions regarding the extend of this work, comparisons including newer models add quality to the submitted work.
> > The main issue is the quality of the image overlay in simulation, which was pointed out by the other reviewers as well.
> > Seeing their positive position regarding this submission, and understanding the scarcity of data resources targeting the flickering removal task, I will increase my rating.

---

> > > ### Author Response · Authors · 2025-08-05
> > >
> > > Dear Reviewer euf9,
> > >
> > > Thank you for your positive feedback and for providing these valuable suggestions for comparison.
> > > We will incorporate your suggestions in the next version of our work, and we hope this research can contribute meaningfully to the study of flicker removal.
> > >
> > > Best wishes,
> > >
> > > Authors of paper 862

---

### Official Review · Reviewer_wLjx · 2025-07-02

**Rating:** 5
**Confidence:** 4

**Summary:**

To address the lack of large-scale realistic flicker removal datasets, the authors propose a robust and scalable benchmark named BurstDeflicker. The benchmark consists of three main components: (1) a Retinex-based synthetic pipeline to simulate realistic flicker effects; (2) the collection of 4,000 real-world flicker-affected images from diverse scenes; and (3) a green-screen-based strategy that introduces motion into image pairs while preserving authentic flicker degradation.

**Dataset Code Accessibility:**

Yes

**Dataset Code Comments:**

The code and dataset are available in the supplementary materials.

**Ethical Considerations:**

No, there are no or only very minor ethics concerns

**Final Justification:**

The author has mostly addressed my concerns. After reviewing the feedback from other reviewers and the author's responses, I have decided to maintain my current rating. However, I suggest that the author conduct a larger-scale user study in the final version, ideally with more than 50 participants, as 10 participants is too few.

**Limitations Weaknesses:**

1. It is unclear how the authors handle the estimation of $\hat{L}_f$ in Equation (3) when generating the synthetic datasets.

2. When using green-screen data to synthesize flicker in dynamic scenes, it is assumed that both the background and the foreground should exhibit flicker. However, the paper lacks specific details on how flicker is applied to the extracted foreground, which may affect the realism of the synthesized flicker.

3. Although the dataset emphasizes scene diversity, it lacks a comprehensive statistical analysis. Important attributes such as motion types, lighting conditions, indoor vs. outdoor environments, and day vs. night scenes are not sufficiently discussed. Such a distributional analysis is essential for evaluating the representativeness of the dataset.

4. The current evaluation of dynamic scenes relies solely on no-reference quality metrics. Incorporating a user study or subjective evaluation would significantly strengthen the authors' claims, as human perception is a critical factor in assessing flicker removal performance.

5. The paper primarily focuses on multi-frame inputs captured under uniform exposure settings. However, flicker artifacts are also prevalent in HDR imaging scenarios. It would be valuable for the authors to discuss the applicability of their dataset and methodology to HDR-related tasks.

**Strengths Contributions:**

1. The paper is well-written and clearly organized, making it easy to follow.
2. The motivation is solid, and the proposed methodology appears to be both novel and reasonable.

---

> ### Author Rebuttal · Authors · 2025-07-30
>
> Thank you very much for your high recognition of our work and your valuable suggestions. We have addressed your concerns as below.
>
> > *It is unclear how the authors handle the estimation of in Equation (3) when generating the synthetic datasets.*
>
> The ratio of $L_f$ to its corresponding effective value $\bar{L_f}$ in Equation (3) defines a general solution form. Specific solutions for $L_f$ including full-wave rectification, half-wave rectification, and PWM, which are derived in Equation (4) using $L_{f\text{-}full}$, $L_{f\text{-}half}$, and $L_{f\text{-}pwm}$, respectively.
>
> > *When using green-screen data to synthesize flicker in dynamic scenes, it is assumed that both the background and the foreground should exhibit flicker. However, the paper lacks specific details on how flicker is applied to the extracted foreground, which may affect the realism of the synthesized flicker.*
>
> Due to the lack of local motion, we composited the motion sequences onto the real burst images using Photoshop (Section 1 in Supplementary Material), preserving the genuine flickering background.
> Currently, we cannot simulate accurate flicker effects on the subject's body. It remains highly challenging to obtain motion sequences with genuine flicker effects **across the entire image**, and this will become a key research focus in the future.
>
> Although the motion in green-screen data is not yet perfect, it can effectively help suppress the occurrence of ghosting phenomena, as shown in Table 2 and Figure 6.
>
> > *Although the dataset emphasizes scene diversity, it lacks a comprehensive statistical analysis. Important attributes such as motion types, lighting conditions, indoor vs. outdoor environments, and day vs. night scenes are not sufficiently discussed. Such a distributional analysis is essential for evaluating the representativeness of the dataset.*
>
> Thank you for the valuable suggestion. We have conducted a data analysis of the dataset, as summarized in the table below.
> |        | Motion types | Light shapes | Indoor: outdoor |  Day: night  | Full-wave: half-wave: PWM | AC-power light intensity (low: medium: high) |
> |--------|:------------:|:------------:|:---------------:|:------------:|:-------------------------:|:-------------------------------------------:|
> | Number |      29      |       4      |   86.5% : 13.5%  | 44.5% : 55.5% |     20% : 47.5% : 32.5%    |                22% : 43% : 35%               |
>
> > *The current evaluation of dynamic scenes relies solely on no-reference quality metrics. Incorporating a user study or subjective evaluation would significantly strengthen the authors' claims, as human perception is a critical factor in assessing flicker removal performance.*
>
> Thank you for your constructive suggestion. In response, we conducted a user study involving 10 participants. Each participant was asked to rate the restoration results on a scale from 1 to 5, where the input image served as the reference for a score of 1, and the ground truth (GT) image corresponded to a score of 5. Higher scores indicate better flicker removal performance and fewer motion-induced ghosting artifacts.
> The results are presented below (with the experimental setup consistent with Table 1 in the main text).
>
> |       | Lin et al. | Retinexformer | Burstormer | HDRTransformer | Restormer |
> | ----- |:----------:|:-------------:|:----------:|:--------------:|:---------:|
> | Score | 2.71       | 3.54          | 3.75       | 3.92           | 4.33      |
>
> > *The paper primarily focuses on multi-frame inputs captured under uniform exposure settings. However, flicker artifacts are also prevalent in HDR imaging scenarios. It would be valuable for the authors to discuss the applicability of their dataset and methodology to HDR-related tasks.*
>
> Thank you for your highly professional advice. When capturing HDR images, different exposures are used, and the short exposure sometimes carries flicker degradation, which can lead to more severe artifacts if forcibly combined.
> Therefore, when we use it in industrial scenes recently, we perform a flicker removal on the short exposure frames before the HDR combination. We will add this discussion and the visual results in the revised version.

---

### Note · Authors · 2025-08-12

Dear Reviewers and Area Chairs,

We sincerely appreciate your valuable time and reviewers' constructive feedback, which have enabled us to substantially improve both the quality of our paper and the proposed dataset. In the rebuttal process, we have addressed all the issues raised during the review and discussion process. Our main improvements are as follows:

* We have provided additional details on the dataset distribution, including lighting types, motion types, illumination intensities, and more.

* We have built a more comprehensive benchmark for state-of-the-art deep learning–based image restoration methods on our dataset.

* We have conducted an extensive user study to complement the quantitative evaluation metrics.

* We have added further ablation studies on the number of input frames.

These revisions have received acknowledgment from all reviewers during the discussion phase. We are grateful to the reviewers wLjx and sgXN who **maintained the positive assessments**, and Reviewer euf9 who chose to **raise the score after the discussion**.

As the first real flicker dataset, we hope that our joint efforts with the reviewers will make a meaningful contribution to the research community. Thank you again for all the efforts that you have made.

Best regards,

Authors of Paper 862

---

### Decision · Program_Chairs · 2025-09-18

**Decision:**

Accept (poster)

**Comment:**

This paper introduces BurstDeflicker, the first large-scale dataset and benchmark for multi-frame flicker removal. The dataset combines synthetic, real-world, and green-screen–based dynamic data, and is accompanied by comprehensive benchmarking on state-of-the-art image restoration models. The contribution is novel, clearly within the scope of the Datasets and Benchmarks track, and addresses an important yet underexplored problem.
While reviewers noted some limitations—such as the relatively small scale of real-image subsets, potential domain gaps in green-screen compositing, and the need for richer evaluation—the authors effectively addressed these through the rebuttal, adding dataset statistics, further benchmarks, ablations, and a user study. Reviewers acknowledged these improvements, with one raising their score. Overall, the work makes a timely and valuable contribution that is expected to serve as a foundation for future research.